Advanced clustering and transfer learning based approach for rice leaf disease segmentation and classification

Yousafzai Samia Nawaz 1 samia@crair.eu
Al-Wesabi Fahd N. 2
Alsolai Hadeel 3
http://orcid.org/0000-0003-1043-2774 Ebad Shouki A. 4
Nasir Inzamam Mashood 5
Fadhal Emad 6
http://orcid.org/0000-0003-1054-4813 Thaljaoui Adel 7
1 Center of Real-World AI Research , Kaunas , Lithuania
2 Department of Computer Science, Applied College at Mahayil, King Khalid University , Mahayil , Saudi Arabia
3 Department of Information Systems, College of Computer and Information Sciences, Princess Nourah Bint Abdulrahman University , Riyadh , Saudi Arabia
4 Center for Scientific Research and Entrepreneurship, Northern Border University , Arar , Saudi Arabia
5 Faculty of Informatics, Kaunas University of Technology , Kaunas , Lithuania
6 Department of Mathematics & Statistics, College of Science, King Faisal University , Al-Ahsa , Saudi Arabia
7 Department of Computer Science and Information College of Science Zulfi, Majmaah University , Al-Majmaah , Saudi Arabia
Sergi Consolato
Electronic publication date: 2025 Jul 28
Publication date: 2025
Volume: 11
Electronic Location ID: e3018
Received 2025 Feb 10; Accepted 2025 Jun 19
Copyright: © 2025 Yousafzai et al.
Copyright year: 2025
Copyright holder: Yousafzai et al.
License: This is an open access article distributed under the terms of the Creative Commons Attribution License, which permits unrestricted use, distribution, reproduction and adaptation in any medium and for any purpose provided that it is properly attributed. For attribution, the original author(s), title, publication source (PeerJ Computer Science) and either DOI or URL of the article must be cited.
License URL: https://creativecommons.org/licenses/by/4.0/

Keywords: Deep transfer learning, Rice leaf disease classification, Segmentation, Contrast enhancement, EfficientNetB0, Feature optimization

Funding: Deanship of Research and Graduate Studies at King Khalid University RGP2/199/46 Princess Nourah bint Abdulrahman University Researchers Supporting Project PNURSP2025R303 Princess Nourah bint Abdulrahman University Researchers Supporting Project Deanship of Scientific Research at Northern Border University, Arar, KSA NBU-FFR-2025 Deanship of Scientific Research, Vice Presidency for Graduate Studies and Scientific Research, King Faisal University, Saudi Arabia KFU252338 This research was supported by the Deanship of Research and Graduate Studies at King Khalid University through the Large Research Project under grant number RGP2/199/46. Princess Nourah bint Abdulrahman University Researchers Supporting Project number (PNURSP2025R303), Princess Nourah bint Abdulrahman University, Riyadh, Saudi Arabia. This research was also supported by the Deanship of Scientific Research at Northern Border University, Arar, KSA through the project number NBU-FFR-2025. This work was supported by the Deanship of Scientific Research, Vice Presidency for Graduate Studies and Scientific Research, King Faisal University, Saudi Arabia (Grant No. KFU252338). The funders had no role in study design, data collection and analysis, decision to publish, or preparation of the manuscript.

==============================
Rice, the world’s most important food crop, requires an early and accurate identification of the diseases that infect rice panicles and leaves to increase production and reduce losses. Most conventional methods of diagnosing diseases involve the use of manual instruments, which are ineffective, imprecise, and time-consuming. In light of such drawbacks, this article introduces an improved deep learning and transfer learning method for diagnosing and categorizing rice leaf diseases proficiently. First, all input images are preprocessed; the images are resized to a fixed size before applying a sophisticated contrast enhanced adaptive histogram equalization procedure. Diseased regions are then segmented through the developed gravity weighted kernelised density clustering algorithm. In terms of feature extraction, EfficientNetB0 is fine-tuned by subtracting the last fully connected layers, and the classification is conducted with the new fully connected layers. Also, the tent chaotic particle snow ablation optimizer is added into the learning process in order to improve the learning process and shorten the time of convergence. The performance of the proposed framework was tested on two benchmark datasets and presented accuracy results of 98.87% and 97.54%, respectively. Comparisons of the proposed method with six fine-tuned models show the performance advantage and validity of the proposed method.

Introduction

Rice is consumed by more than fifty percent of the world’s population, and any harm to the crops leads to losses in business for farmers around the globe (Asibi, Chai & Coulter, 2019). Such losses affect the lives of farmers, food security, poverty, and economic growth, besides being a challenge to attaining sustainable development. An investigation by the International Rice Research Institute (IRRI) revealed that rice diseases can potentially slash yields by 80% cutting farmers’ profits, food availability, and consumer costs (Shew et al., 2019). It leads to forcing farmers to use pesticides and chemicals that, in one way or another, have negative impacts on human health and the environment (Jones, 2021). Hence, the control of rice diseases affecting the leaves is crucial to promoting sustainable rice production, and hence food security in the world (Tudi et al., 2021; Damalas & Koutroubas, 2016). Over the last few years, rice classification and quality detection have emerged as applicable requirements since this food staple is a lifeline for millions of food-insecure consumers (Tang et al., 2022). Earlier, diseases were detected through manual identification and expert systems, both of which are slow and inaccurate (Yan et al., 2020; Yousafzai et al., 2025).

Among these factors, leaf diseases have a significant impact on a country’s rice production because they are not frequently checked. Many farmers may not know some of these diseases and the time they occur, thereby leading to occasional outbreaks on the plants, causing the yields of rice to reduce on aggregate. Classically, plant disease diagnosis is a process where an expert observes that there is a slight difference in the color of the leaves, which is time-consuming and requires manual work. This approach also has a significant problem in determining the level of harm or intensity on large-scale farms. Early disease diagnosis is a very key success factor since both the production quantities and the quality of rice are affected by diseases. When diseases in plants are detected early, appropriate measures are taken to avoid disease transfer and ensure the plants yield healthier crops, hence enhancing rice yield and availability in supply (Bari et al., 2021). Consequently, the search for adequate means of early identification of plant diseases has turned into one of the crucial requirements of contemporary agriculture.

Over the past few years, AI-based methodologies, such as machine learning (ML) and deep learning (DL) have been identified to be valuable tools in the automatic identification in various domains including healthcare (Nasir, Alrasheedi & Alreshidi, 2024; Nasir et al., 2024) language processing (Yousafzai et al., 2024; Alzaidi et al., 2024) and agriculture (Zheng et al., 2019). Different research has pointed to the use of ML approaches in detecting crop leaf attacks; however, these approaches lead to higher errors when working with larger images of the crop leaves and cause more time spent in training. Therefore, to detect diseases in crops, DL methods have been investigated since they outperform other methods in handling large databases with fewer features. Convolutional neural networks (CNNs) have become important in the development of ML for agriculture applications owing to their ability to extract virtually all data features. Based on the performance profile of CNNs and the current status of datasets and automated algorithms for rice leaf disease identification. The proposed study examines various CNN-based deep transfer learning models for the diagnosis of rice leaf diseases. The following is a comprehensive representation of the primary contributions of the proposed works: Contrast enhanced adaptive histogram equalization (CEAHE) is suggested as a preprocessing method to enhance the contrast characteristics of the rice leaf images.

For segmenting the diseased portions in the images, an improved density clustering method called gravity weighted kernelized density clustering (GWKDC) is employed.

The study adopts transfer learning in the construction of a fine-tuned EfficientNetB0 to extract deep features, and a fully connected layer is utilized for the classification.

A comparative analysis is then performed to compare the proposed model with six deep transfer learning models to check the performance.

This study provides a tent chaotic particle swarm with a snow ablation optimizer (TCPSAO) algorithm proven to establish the most accurate learning rate and enhance the execution of the learning rate optimizer and the model training.

The subsequent sections of the article are organized as follows: “Literature Review” describes the literature related to this article. “Proposed Methodology” outlines the proposed framework. “Experimental Results” details the results, expressed as numerical values. Ultimately, “Conclusion” concludes the work.

Literature review

Based on the different types of knowledge available, we can understand that the research community is interested in the subject of automated disease detection using DL approaches (Li, Zhang & Wang, 2021). CNN and support vector machine (SVM) approaches are utilized for rice disease classification. An additional pooling layer of this module, changes in the activation process, they reported to achieve an average accuracy of 96.8% on a self-generated dataset (Jiang et al., 2020). In the same way, a two-layer detection model is suggested using region-based convolutional neural network (RCNN) to detect Brown Rice Planthoppers in photos with thorough results of 94.5% accuracy rate and 88% recall rate. This study also compared their outcome to that of the YOLO v3 algorithm for comparison (He et al., 2020). In another study, a model to diagnose rice leaf diseases that involves bacterial blight, leaf spot, and brown spot with the help of hue threshold segmentation, this model integrates a gradient boosting decision tree to improve the result with an accuracy of 86.58% (Azim et al., 2021).

Furthermore, a technique using deep convolutional neural networks (DCNN) was presented to detect rice diseases and use CNNs trained for the identification of ten different disease types of rice leaves with a high accuracy of about 95.48% by applying an independent 10-fold cross-validation (Lu et al., 2017). The simple CNN and InceptionResNetV2 architectures were put forward together with their respective configurations. The residual model used in the research involved the InceptionResNetV2 model, in which transfer learning was used for feature extraction with parameters retrained to perform the classification, getting an accuracy of 95.67% (Krishnamoorthy et al., 2021). In Thepade, Abin & Chauhan (2022), rice leaf segmentation was achieved by Otsu thresholding and block truncation; however, this technique predominantly relies on pixel intensity distributions and exhibits suboptimal performance under fluctuating lighting conditions. Similarly, Hasan et al. (2023) employed K-means clustering in conjunction with a proprietary CNN; however, the model relies on pre-established cluster quantities, rendering it susceptible to outliers and the selection of initial centroids. While deep learning-based segmentation (Lu et al., 2017; Krishnamoorthy et al., 2021) is computationally demanding and often suboptimal for real-time applications, it provides superior accuracy. The research offers an attention-based depthwise separable neural network optimized via Bayesian optimization (ADSNN-BO) that improves MobileNet attention layers to increase feature learning and reduce computational cost. Despite limited training, the model outperformed Xception and DenseNet121 with 94.65% test accuracy. Limitations include the limited dataset (2,370 samples), generalization challenges to varied settings, and validation on larger public datasets (Wang, Wang & Peng, 2022). The SEWA-SPBO Deep Maxout Network, optimized using a hybrid of the sunflower earthworm algorithm and student psychology-based optimization techniques, is utilized for routing and classifier training. black hole entropic fuzzy clustering (BHEFC) segmentation improves input quality. The accuracy was 93.92%, with a sensitivity of 95.76% and specificity of 89.38%; nevertheless, limitations in the dataset, controlled lighting conditions, and scalability challenges in different agricultural contexts impeded its effectiveness (Shanmugam et al., 2023). A detailed literature survey summary is presented in Table S1.

The research provides substantial advancements in the agricultural diagnosis of rice leaf diseases by addressing key constraints of prior studies, including inadequate preprocessing and segmentation precision. Contemporary methodologies—thresholding, clustering, and segmentation utilizing CNN—exhibit a lack of adaptability to fluctuating light conditions, heightened sensitivity to noise, and inadequate edge preservation. Conventional optimizers such as the particle swarm optimization (PSO), genetic algorithm (GA), and snow ablation optimizer (SAO) exhibit disadvantages, including susceptibility to premature convergence and insufficient global search capability. This study introduces a lightweight and efficient system that incorporates graph weighted kernelized density clustering (GWKDC) for precise, edge-sensitive segmentation and tent chaotic PSO–SAO (TCPSAO) for resilient and adaptive optimization. Both components significantly enhance classification performance, as demonstrated by comparison and ablation analyses.

Proposed methodology

This section presents the proposed framework for the classification of rice leaf disease using images. The architecture of this rice disease classification is described in Fig. 1. As presented by this figure, in the experimentation, two datasets are used for which the data are freely available in the public domain.

Figure 1 Architecture of proposed framework.

Dataset

For evaluation, we used the dataset provided by Sethy et al. (2020) in their work about disease diagnosis for rice leaves employing an SVM with deep feature extraction. This rice leaf dataset initially consisted of 5,932 images with four classes of rice diseases. To that, 1,308 images of Tungro, 1,440 images of Blast, 1,584 samples of Bacterial Blight, and 1,600 images of Brown Spot were incorporated. A total of 70% of the dataset was utilized for training, 10% for validation, and the remaining 20% was used for testing. This dataset is published in Mendeley Data, which is helpful for DL/ML studies focused on enhancing rice disease identification.

In this experiment, the Kaggle Rice Leaf Dataset (https://www.kaggle.com/datasets/dedeikhsandwisaputra/rice-leafs-disease-dataset, accessed on 30 June 2024) is utilized which comprises a total of 2,627 images of rice leaves diagnosed as having five specific diseases namely bacterial leaf blight, brown spot, leaf blast, leaf scald, narrow brown spot, and a normal one—healthy leaves. A total of 1,840 images were allocated for training, 525 for testing, and 262 for validation, derived from a distribution of 70% for training, 20% for testing, and 10% for validation. Each category contained a comparable quantity of samples—307 images for training, 87 images for testing, and 43 images for validation—to ensure class balance. The sample images of both datasets are provided in Figs. S1 and S2.

Preprocessing

The images in the raw dataset possess lower contrast than those originating from the raw dataset, affecting the accuracy during the training phase. For this reason, a preprocessing stage is incorporated, and it aims to contrast and enhance the recorded rice leaf images. These include, for instance, resizing the image and improving the colors. First, each input image of a rice leaf is resized to have dimensions of 380 × 380 pixels, allowing the application of the CEAHE algorithm to enhance image quality. Consequently, in CEAHE, the images are divided into subregions, and the global entropy of the input image is employed to determine the extent of each subregion. The mathematical expression for the entropy E of the contrast-enhanced image is as follows:

(1) E=−∑x=0255Pxlog2(Px)

In this regard, Px denotes the probability associated with the intensity level x. The exponential decay function (EDF) is subsequently established using the derived entropy E and the image size N. The EDF can be represented numerically as:

(2) ϕ=(N−Emaxexp⁡(−μE)max).

Taking into account the Emax value of the contrast-enhanced image is determined as 8. For this Emax, the EDF analyses the image pixels and comes with a result that 8×8 pixels should suffice. However, if the entropy is lower than the mentioned, for instance, if Entropy=1, the pixel size is shrunk to half of the original.

Segmentation

The preprocessed images are subsequently input into the segmentation component for enhanced detection of the diseased region. To this effect, this research proposes the method GWKDC to facilitate the efficient segmentation of the salient regions of rice leaf images. This technique shows high accuracy in identifying regions of interest for better and more efficient overall segmentation. Algorithm 1 presents the step-by-step process of the proposed segmentation method.

Algorithm 1 Gravity weighted kernelized density clustering (GWKDC).

Input: Image X∈RA×B, bandwidths B1, B2, polynomial degree n	
Output: Segmented mask M	
Convert image X to grayscale if necessary;	
Fit a polynomial of degree n to each row or column of X;	
Estimate coefficients p0 to pn using least squares method;	
Compute the first derivative dydr and find peak points;	
Validate peaks using second derivative condition d2ydr2<0;	
Use valid peaks as initial cluster centers;	
foreach pixel pair (v,w) do	
  Compute gravity kernel distance:	
   Gvw=exp⁡(−(rv−yv)2+(rw−yw)2B1)⋅exp⁡(−(Xv−Xw)2B2);	
Assign each pixel to the cluster with minimum Gvw;	
Optionally merge overlapping clusters based on centroid distance;	
Generate segmentation mask M from final clusters;	
return M	

By examining the extracted features from the images, it is demonstrated that diseased leaf images can be divided into two main components: One part is the lesion, the entire leaf, and extraneous background areas, while the other part is the pathology and the regions of the leaf with the disease. The pixel values of the image cover only 0 to 255, and hence it becomes challenging to situate the peak point because of local internal points. To address this complexity, the proposed method applies sub-polynomials to detect the peak points, which can be expressed mathematically as:

(3) y=p0+p1r+p2r2+⋯+pnrn.

Here, r stands for the differentiation parameter of the pixel values of the image input, and y is the number of pixels in an image. The training data can be represented in the form {ra,ya∣a=0,1,2,…,255} for M = 255; where for every training input r there is the training output y. The model is constructed in such a way that M is greater than n. Therefore, Eq. (3) can be mathematically defined as:

(4) y=p0+p1rk+p2rk2+⋯+pnrkn,k=0,1,…,255.

The coefficients p0,p1,…,pn are determined using the Least Squares (LS) method to minimize the squared error between the fitted polynomial and actual data values. The total squared error function is given by:

(5) S=∑k=0M−1(yk−p0+p1rk+p2rk2+⋯+pnrkn)2.

To find optimal coefficients, partial derivatives of S with respect to each pj are computed and set to zero, resulting in a solvable system of equations.

(6) ∂S∂p0=∑k=0M−1(yk−(p0+p1rk+p2rk2+⋯+pnrkn))2=0,∂S∂p1=∑k=0M−1(yk−(p0+p1rk+p2rk2+⋯+pnrkn))2=0,∂S∂pn=∑k=0M−1(yk−(p0+p1rk+p2rk2+⋯+pnrkn))2=0.

By taking the derivative of Eq. (3) and identifying the peak points, solve the first derivative of the polynomial to locate critical points. This makes it possible to find the maximum or critical point of the data. The resulting expression after applying the derivative can be mathematically represented as:

(7) dydr=p1+2p2r+⋯+npnrn−1=0.

Two conditions must be met accurately to determine the peak points. First, for the polynomial taken, the derivative must be equal to zero to show that there can be a peak. Second, the polynomial value at this zero point needs to be greater than the values at the nearby points to ensure one is at the peak. These conditions guarantee that the peak is not only a relative maximum, but it is the maximum among all values in the immediate neighborhood of its location.

After all the peak points for the features are found, the peak points of diseased regions of the given plant leaf are selected randomly. These peak points are then considered cluster centers, and then the diseased leaf areas can be grouped. In this case, the number of clusters is determined based on the peak points in a pre-defined conventionalized manner, together with the determination of the respective centers of the clusters.

The resolution of the image is assessed prior to determining the distance between two pixels. Which is done by assessing the total number of pixels, represented by the parameter mk{k=0,1,2,…,255}, having different pixel intensity frequency distribution. The pixel quality measure Qk is defined based on the relative frequency of pixel intensities:

(8) Qk=mk⋅1∑k=0255mk,k=0,1,2,…,255.

Suppose we have an image X with size A × B, and points X(r,y) are arbitrary. Here, r and y represent the pixel intensity levels of an image, which go from 1,2,…,A and 1,2,…,B. Suppose we have two-pixel points, X(rv,yv) and X(rw,yw) and their pixel values are mentioned as Xs and Xw. The kernel gravity distance Gvw between these two-pixel points is expressed as:

(9) Gvw=exp⁡((rv−yv)2+(rw−yw)2B1)exp⁡((Xv−Xw)2B2).

In this context, the spatial kernel gravity distance between two pixels is manipulated by exp⁡((rv−yv)2+(rw−yw)2B1), while the grey kernel gravity distance between two pixels is manipulated by exp⁡((Xv−Xw)2B2). The bandwidth of the kernel gravity density function is interpreted by B1 and B2. Ultimately, the image pixels are used to accurately cluster and segment the afflicted regions. Figure 2 presents some samples of preprocessed and segmented images.

Figure 2 Sample images taken after preprocessing and segmentation.

Pretrained EfficientNetB0 model for feature extraction

EfficientNet uses compound scaling—that is, a single coefficient—to equally scale width, depth, and resolution and preserve proportionate model growth as input size grows. Based on MobileNetV2, EfficientNet-B0 combines inverted bottleneck blocks for rapid processing at high resolution with squeeze-and-excitation (see Fig. S3).

Pretrained on ImageNet, EfficientNet-B0 was retrained with a learning rate of 0.005, momentum of 0.703, and 100 epochs using stochastic gradient descent by replacing its last layers. Particularly helpful in fields with little labeled data like agriculture, transfer learning lets one effectively adapt to the rice disease dataset.

(10) fθ=(Cθ,Lθ).

In this case, signifies Lθ∈Is1 the label, while Cθ∈I∂Ws1 represents a d-dimensional feature space. Here, s1 denotes the total count of source samples. The “target” dataset is indicated by fT=CT, where CT∈I∂Ws2 and s2 refers to the total number of target samples.

The structure of the ‘domain’ f={C,P(i)} of feature data C and its marginal probability distribution P(i) in the range of [0,1]. These can be equated to labels L and function j(i), anticipating what the tasks are and giving them the label T={L,j(i)}. In general, the outlined process is depicted in Fig. 3.

Figure 3 Visual representation of deep transfer learning process.

Following the training of the DL models orchestrated by deep transfer learning, deep features were obtained. The dataset employed for this study includes the Kaggle and Mendeley rice disease datasets; therefore, deep features were extracted for both datasets. The features were extracted from the Global Average Pooling layer of every trained model and passed to a fully connected layer for classification.

Feature optimization

SAO is a nature-inspired algorithm based on melting and sublimation dynamics, effective in addressing population diversity and convergence variability. While SAO performs well in identifying local optima, it struggles to scale globally and often gets trapped in local solutions, resulting in longer training times. To overcome these issues, we propose a combinatorial enhancement termed particle swarm with snow ablation optimizer (PSAO). Moreover, since SAO uses a randomized initialization method that may limit the search space due to uneven solution distribution, we incorporate tent chaotic mapping to generate a more diverse and uniformly distributed initial population. This chaotic mechanism prevents premature convergence and improves global search capabilities (Vaswani et al., 2021). The resulting hybrid algorithm, TCPSAO, further improves exploration by combining chaotic initialization with PSO-enhanced position updates. This adjustment strengthens the algorithm’s global search ability and helps escape local optima. The position update mechanism of TCPSAO is defined in Eqs. (11) and (12), where RB(h) is a Brownian motion-based Gaussian vector, ⊗ denotes element-wise multiplication, r1∈[0,1], r2∈[−1,1], and X is the population centroid as defined in Eq. (17).

(11) xi(h+1)={Elite(h)+RBi(h)⊗(r1×(Oi(h)−Xi(h))+(1−r1)×(X¯−Xi(h))),M×Oi(h)+RBi(h)⊗(r2×(Oi(h)−Xi(h))+(1−r2)×(X¯−Xi(h)))

(12) Xi(h+1)=xi(h+1)+Vi(h+1)

(13) Elite(h)∈[O(h),Xsecond(h),Xthird(h),XC(h)].

In Elite(h), the population is constituted by different elite subgroups, and the subjects are selected randomly as shown in Eq. (13). O(h) represents the current best, Xsecond represents the second best, and Xthird(h) represents the third-best one in the current population. Further, XC(h) describes the centroid position estimated by the fitness ranks of 50% of the top performers among the group.

(14) XC(h)=1N1∑i=1N1Xi(h).

where N1 is the number of leaders and N1 = (1/2) * total population size. Xi(h) is the rank with of a top-performance leader in the i-th iteration of the algorithm.

(15) M=(0.35+0.25×eh/hmax−1e−1)×H(h),H(h)=e−h/hmax.

Here, variable M stands for the melting rate, which is one of the main parameters that controls the melting dynamics during the development phase, as described by Eq. (15). The outside condition for termination is indicated by hmax. At the same time, H(h) is the average temperature h.

(16) FRB(x;0,1)=12π×exp⁡(−x22)

(17) X¯(h)=1N∑i=1NXi(h).

The two-population mechanism is established, and the total population and the two subpopulations are identified as N, Na, and Nb, respectively. The subpopulation Na is assigned to engage in exploration tasks more frequently than the subpopulation Nb, which is liable for exploitation tasks. Here, it should be stated that, during the iterations, the size of Nb is gradually reduced, which creates a possibility of increasing the size of Na in an appropriate measure. The variables index a and index b are vectors, which are the sets of indexes corresponding to the line numbers of individuals in Na and Nb, the incomplete position matrices.

In TCPSAO, the iterative process starts with generating a population randomly by using the following formulas that produce N×dim matrix as shown in Eq. (18). After this, the initial population solution is improved by combining it with tent chaotic map which is expressed as Eq. (19). This step helps in creating a more diversified and well distributed initial solution. The last of the initial population, given the tent chaotic map, is found according to Eq. (20). Then, the assigned positions, velocities, best fitness values, objective functions, elite members, the global best position, best position, and the global best fitness of the swarm are defined.

(18) x=lb+rand(N,dim)×(ub−lb)

(19) F(x,α)={2xα,0≤x<α,2(1-x)1-α,α≤x≤1.

(20) xi+1,g=F(xi,g,a),i=1,2,…,N

(21) X=xi+1,g×ub.

lb indicates the lower bound of the population, N denotes the population size, g and dim signify the representative dimensional size, α represents the parameter of the tent chaos map, and ub denotes the upper bound of the population.

Experimental results

The Kaggle and Mendeley rice leaf datasets were utilized independently for training and testing to assess the proposed model’s generalizability across different domains. The two datasets relate to rice leaf illnesses but vary in disease class representation, image resolution, lighting conditions, and visual characteristics. Moreover, many classes, such as “Brown Spot” and “Bacterial Leaf Blight,” appear in both databases but possess distinct annotations or are documented under varying field conditions. Integrating the datasets may introduce label noise, class redundancy, and skewness, thereby undermining the learning process. Independently testing the model on both datasets provides a more robust evaluation of its resilience to data distribution alterations and facilitates a more authentic testing environment for real-world deployment. Thus, MobileNetv3, InceptionResNetv2, NasNetLarge, Xception, EfficientNetB2, AlexNet, and EfficientNetB0 were utilized for feature extraction, and the classification was performed with a fully connected layer. The evaluation process comprises three steps: (i) baseline performance utilizing deep transfer learning without segmentation or optimization; (ii) performance subsequent to the integration of the proposed TCPSAO; and (iii) final performance following the incorporation of both segmentation and optimization. This approach facilitates a systematic evaluation of each activity in relation to the model’s overall enhancement. All experiments were conducted in Python on a desktop with 16 GB RAM and an 8 GB graphics card.

Mendeley rice leaf disease results

Table 1 shows a quantitative analysis for several DL models using the Mendeley rice leaf disease dataset, unsegmented, and unoptimized. All the variations of EfficientNet offered decent accuracy with EfficientNetB0 giving the highest efficiency; the highest accuracy was 92.45%, precision of 92.32%, recall of 92.07%, F1-score of 92.58%, and an area under the curve (AUC) of 95.45%. Much the same, the EfficientNetB2 achieved different success with 90.76% accuracy and 89.33% F1-score. On the other hand, AlexNet possessed the lowest efficacy of 64.39% as well as an F1-score of 60.97% only. Other models, including MobileNetv3 and InceptionResNetv2, had pretty good performance, with accuracies of 82.43% and 80.38%, respectively.

Table 1 Performance comparison of pre-trained deep learning models on the Mendeley dataset using raw (unsegmented and unoptimized) images.

This table establishes the baseline performance for each model.

Model	Precision (%)	Recall (%)	F1-score (%)	Accuracy (%)	AUC (%)	
MobileNetv3	81.36	82.12	81.05	82.43	88.91	
InceptionResNetv2	80.18	80.10	79.89	80.38	85.26	
NasNetLarge	76.15	77.34	76.93	79.54	85.38	
Xception	81.43	82.57	81.21	82.32	87.51	
EfficientNetB2	89.08	90.21	89.33	90.76	95.49	
AlexNet	61.51	62.43	60.97	64.39	70.76	
EfficientNetB0	92.32	92.07	92.58	92.45	95.45	

Table 2 compares the performances of deep transfer learning models on the Mendeley rice leaf disease dataset using preprocessed images with TCPSAO optimization. However, segmentation was not applied. The best performance came from the EfficientNetB0 model with an accuracy of 95.71% for classification accuracy, a precision of 94.76%, a recall of 95.03%, an F1-score of 94.98%, and an AUC of 97.43%, indicating its ability to diagnose rice diseases better. MobileNetv3 had almost similar performance with an accuracy of 91.76%, and an F1-score of 90.33% while NasNetLarge achieved an accuracy of 86.19% and InceptionResNetv2 achieved 85.32% accuracy overall. In contrast, for the AlexNet model, the lowest model accuracy of 66.67% and F1-score of 63.01% was obtained.

Table 2 Performance of the same pre-trained models on the Mendeley dataset using preprocessed images with TCPSAO optimization applied, but without segmentation.

This table evaluates the isolated impact of optimization.

Model	Precision (%)	Recall (%)	F1-score (%)	Accuracy (%)	AUC (%)	
MobileNetv3	89.98	90.54	90.33	91.76	94.21	
InceptionResNetv2	84.04	84.67	84.25	85.32	86.46	
NasNetLarge	85.32	85.33	85.17	86.19	89.11	
Xception	82.87	83.21	83.44	84.23	87.78	
EfficientNetB2	86.71	87.24	86.93	88.16	91.27	
AlexNet	63.56	63.98	63.01	66.67	72.67	
EfficientNetB0	94.76	95.03	94.98	95.71	97.43	

Table 3 shows the comparison of DL models after preprocessing, segmentation, and optimization done on the Mendeley rice leaf disease dataset. The EfficientNetB0 received the highest score in all measurements, with 98.87% of accuracy, 98.42% of precision, 97.39% of recall, and a nearly perfect AUC of 99.87%, which makes it the best model for this task. EfficientNetB2 also demonstrated a good result with the accuracy of the model of 95.05% and a relatively high F1-score of 94.32%. MobileNetv3 is next with an accuracy of 93.45% and an F1-score of 91.72%. The rest of the models delivered reasonable performances, getting accuracies between 88.42% and 91.87% with other models like InceptionResNetv2, NasNetLarge, and Xception. Overall, however, AlexNet recorded the lowest accuracy of 68.35% and AUC of 71.76% making it less capable of managing the current dataset efficiently. Figure S4 shows training and validation accuracy as well as training and validation graphs for the proposed model on the Mendeley dataset.

Table 3 Classification results for all models on the Mendeley dataset after applying both segmentation (GWKDC) and TCPSAO optimization.

This demonstrates the complete pipeline’s effect on model performance.

Model	Precision (%)	Recall (%)	F1-score (%)	Accuracy (%)	AUC (%)	
MobileNetv3	92.55	89.25	91.72	93.45	96.81	
InceptionResNetv2	90.67	90.93	91.11	91.87	92.57	
NasNetLarge	86.24	87.36	87.96	88.42	93.34	
Xception	87.51	88.95	88.33	88.75	95.78	
EfficientNetB2	93.84	91.50	94.32	95.05	95.43	
AlexNet	65.87	66.85	65.75	68.35	71.76	
EfficientNetB0	98.42	97.39	97.48	98.87	99.87	

Table 4 compares the performance of TCPSAO with SAO and PSO optimization techniques. The accuracy is 98.87%, precision 98.42%, and AUC 99.87% for TCPSAO, which is considered the best method in terms of both precision and generalization, and TCPSAO’s performance is further enhanced when the chaotic map is applied to improve exploration and optimization. SAO, while slightly lagging behind TCPSAO, is also highly efficient with an accuracy of 95.31%, the precision of the results being 94.31%, as well as an AUC of 96.98%, which proves that despite the optimization problems it solves, it does not possess that superior global search that TCPSAO possesses. Although reasonable, the results achieved by PSO are lower compared to the results of the other two techniques: accuracy of 92.67% and AUC of 94.54%, which shows the worst efficiency of the algorithm in the conditions of the complex optimization. Table 5 clearly demonstrates that TCPSAO significantly outperforms conventional optimization methods such as PSO and SAO, hence substantiating its efficacy in enhancing feature selection and convergence.

Table 4 Comparison of various optimization techniques on the Mendeley dataset.

Optimizer	Precision (%)	Recall (%)	F1-score (%)	Accuracy (%)	AUC (%)	
SAO	94.31	94.56	94.11	95.31	96.98	
PSO	89.98	91.32	90.89	92.67	94.54	
TCPSAO	98.42	97.39	97.48	98.87	99.87	

Table 5 Baseline classification performance of deep learning models on the Kaggle rice disease dataset using raw images (no segmentation or optimization).

Model	Precision (%)	Recall (%)	F1-score (%)	Accuracy (%)	AUC (%)	
MobileNetv3	80.24	81.18	80.12	83.15	87.54	
InceptionResNetv2	87.27	87.50	87.38	88.12	91.76	
NasNetLarge	80.55	80.36	80.46	80.72	87.43	
Xception	75.64	76.10	75.85	77.34	83.78	
EfficientNetB2	86.34	86.17	86.19	87.18	87.54	
AlexNet	70.24	70.18	70.21	70.12	74.21	
EfficientNetB0	91.87	92.51	91.65	93.74	96.87	

Kaggle rice leaf disease results

Table 5 shows the results of different convolutional networks and architectures utilized in the Kaggle rice leaf disease picture dataset, which does not include segmentation or optimization. EfficientNetB0 has the highest accuracy of 93.74%, the highest precision of 91.87% on average and 92.51% recall proving EfficientNetB0 is on top of the model in recognizing diseased rice leaves. InceptionResNetv2 also performs exceptionally well with a test accuracy of 88.12% and a precision and recall rate of approximately 87%. However, both EfficientNetB2 and MobileNetv3 present moderate results obtained with test set accuracies of 87.18% for EfficientNetB2 and 83.15% for MobileNetv3. At the lowest level of optimization, AlexNet has the lowest level of efficiency having a 70.12% accuracy level.

The assessment of a range of deep transfer learning models on the Kaggle rice leaf disease dataset, shown in Table 6, presents significant divergences in the performance indices, all pointing to the progress in the use of technology in identifying diseases affecting crops. Amongst all the models, EfficientNetB0 provides the best results, securing an accuracy of 95.12%, a precision of 94.12%, and a recall of 94.56%. EfficientNetB2 comes right behind with 90.38% accuracy, followed by MobileNetv3 with 90.15% accuracy. Other older architectures, such as AlexNet are far behind at 68.31%.

Table 6 Results of the same models on the Kaggle dataset after applying TCPSAO optimization only.

Model	Precision (%)	Recall (%)	F1-score (%)	Accuracy (%)	AUC (%)	
MobileNetv3	89.89	90.21	89.76	90.15	94.76	
InceptionResNetv2	86.24	86.72	86.48	87.22	90.15	
NasNetLarge	80.87	81.29	80.57	82.11	89.42	
Xception	80.24	80.18	80.21	80.29	87.56	
EfficientNetB2	87.36	89.45	87.72	90.38	95.34	
AlexNet	67.62	68.58	67.34	68.31	77.56	
EfficientNetB0	94.12	94.56	94.65	95.12	98.63	

The comparison in Table 7 presents DL models using the Kaggle rice leaf disease dataset to demonstrate high accuracy, especially with the application of image preprocessing, segmentation, and optimization techniques. Ranking at the top is EfficientNetB0 with an overall accuracy of 97.54%, a precision of 97.09%, and a recall of 97.28%, which indicates it performs promisingly well at accurately differentiating the rice leaf diseases. EfficientNetB2 also turned out rather well and achieved an accuracy of 92.51%. Just as follows are MobileNetv3 and InceptionResNetv2, which achieved net accuracies of 91.15% and 90.12%, respectively. Training and validation curves of the proposed model are illustrated in Figure S5.

Table 7 Final evaluation of models on the Kaggle dataset after applying both GWKDC segmentation and TCPSAO optimization.

Model	Precision (%)	Recall (%)	F1-score (%)	Accuracy (%)	AUC (%)	
MobileNetv3	89.65	90.11	90.64	91.15	94.65	
InceptionResNetv2	89.37	89.42	89.28	90.12	96.12	
NasNetLarge	78.29	79.35	78.03	81.57	87.56	
Xception	84.47	84.21	83.29	85.23	87.29	
EfficientNetB2	90.11	90.12	89.52	92.51	95.60	
AlexNet	69.26	70.64	69.38	71.62	77.19	
EfficientNetB0	97.09	97.28	96.76	97.54	99.48	

The comparison of the optimization techniques shown in Table 8 also focuses on the effectiveness of the opted TCPSAO technique that yields excellent accuracy of 97.54% and precision of 97.09%. Concerning the evaluation of this method, the recall value of 97.28% and an AUC of 99.48% suggest its effectiveness in the recognition of rice leaf diseases across the four measurements. On the same note, the PSO method performs a very impressive score, with an accuracy of 94.41% a precision of 92.89%, and a recall of 93.51%. Though it is equally effective and stable, there is a marked improvement in the results due to TCPSAO. Likewise, an accuracy of 93.70% is obtained by applying the SAO method. The visual comparison analysis of optimizers’ performance on both datasets is illustrated in Figure S6.

Table 8 Comparison of various optimization techniques on the Kaggle dataset.

Optimizer	Precision (%)	Recall (%)	F1-score (%)	Accuracy (%)	AUC (%)	
SAO	93.15	93.29	92.79	93.70	96.58	
PSO	92.89	93.51	92.34	94.41	96.74	
TCPSAO	97.09	97.28	96.76	97.54	99.48	

Segmentation performance analysis

Table 9 illustrates a comparative analysis of the segmentation performance of four methodologies—SegNet, K-means, UNet, and the proposed GWKDC approach—utilizing Mendeley and Kaggle datasets. Each evaluation is conducted using a carefully annotated test set. Metrics assessed include intersection over union (IoU), dice coefficient, precision, and recall. GWKDC surpassed the baseline UNet model on the Mendeley dataset, achieving an IoU of 83.17% and a Dice score of 88.03%, compared to the UNet’s IoU of 81.34% and Dice score of 86.21%. Both conventional segmentation methods, SegNet and K-means, demonstrated significantly inferior outcomes; SegNet achieved 70.45% IoU, whilst K-means attained 70.89%. GWKDC achieved superior segmentation scores of 81.76% IoU and 86.72% Dice, surpassing UNet (79.42% IoU and 84.66% Dice), as well as K-means and SegNet on the Kaggle dataset. GWKDC demonstrates robust and adaptable performance across both datasets in processing rice leaf images under diverse conditions. Consequently, it is an excellent option for downstream sickness classification tasks, since its high recall and accuracy values indicate effective border preservation and a reduced incidence of false positives.

Table 9 Segmentation performance comparison across Mendeley and Kaggle datasets using various methods.

Dataset	Method	IoU (%)	Dice (%)	Precision (%)	Recall (%)	
Mendeley	SegNet	70.45	76.22	73.38	78.14	
K-means	70.89	76.03	73.35	77.98	
UNet	81.34	86.21	84.73	87.85	
GWKDC (Proposed)	83.17	88.03	86.41	89.62	
Kaggle	SegNet	68.39	74.86	71.90	76.72	
K-means	69.11	75.34	72.83	76.89	
UNet	79.42	84.66	83.19	86.07	
GWKDC (Proposed)	81.76	86.72	85.23	88.54	

Computational efficiency analysis

The computational performance of several well-known pre-trained models and the suggested method is thoroughly contrasted in Table 10. The evaluation covers parameters, model disk size, training length, and inference time per image. Combining EfficientNetB0 with the GWKDC segmentation and TCPSAO optimization modules yields the proposed approach that strikes a compromise between computational economy and performance among all models. With an inference time of 0.80 ± 0.05 s per image, the training runs about 66 ± 3 min. Lightweight models such as MobileNetV3 and EfficientNetB0 (baseline) compromise segmentation efficacy and classification accuracy even if they offer accelerated inference. On the other hand, more complicated architectures like NasNetLarge and InceptionResNetV2 show somewhat high training and inference costs, which makes them less suited for real-time or resource-limited environments. The results show that the proposed approach is computationally viable and exact for application in functional agricultural disease detection systems.

Table 10 Computational comparison of selected pre-trained models and the proposed method.

Inference time is reported in seconds per image, and it is averaged over three trials.

Model	Parameters (M)	Model size (MB)	Training time (min)	Inference time (s/image)	
MobileNetV3	5.4	16	39 ± 2	0.62 ± 0.04	
InceptionResNetV2	55.9	215	70 ± 4	0.88 ± 0.06	
NasNetLarge	88.9	343	79 ± 5	1.03 ± 0.07	
Xception	22.9	84	58 ± 3	0.78 ± 0.05	
EfficientNetB2	9.1	90	62 ± 3	0.82 ± 0.05	
AlexNet	61.0	233	52 ± 3	0.91 ± 0.06	
EfficientNetB0 (Baseline)	5.3	29	48 ± 2	0.65 ± 0.04	
Proposed model	6.8	41	66 ± 3	0.80 ± 0.05	

Ablation studies

The unoptimized EfficientNetB0 model without segmentation attains an accuracy of 92.45% on the Mendeley dataset and 93.74% on the Kaggle dataset, as indicated in Table 11. The application of the TCPSAO optimization module results in a 3.26% increase in accuracy on Mendeley, reaching 95.71%, and a 1.38% increase on Kaggle. Similarly, the exclusive utilization of the GWKDC segmentation module enhances Kaggle by 1.93% and Mendeley by 4.38%. The suggested method attains a maximum accuracy of 98.87% on Mendeley, reflecting an increase of 6.42% from the baseline, and 97.54% on Kaggle, indicating a 3.80% improvement when both GWKDC and TCPSAO are integrated. The results validate the efficacy and complementarity of the proposed technique, demonstrating that each module enhances performance separately, while their combined application yields the greatest benefits.

Table 11 Ablation study on the effect of GWKDC segmentation and TCPSAO optimization using EfficientNetB0 on Mendeley and Kaggle datasets.

Configuration	Accuracy (%)—Mendeley	Accuracy (%)—Kaggle	
Baseline EfficientNetB0 (no segmentation/opt)	92.45	93.74	
Baseline + TCPSAO	95.71	95.12	
Baseline + GWKDC	96.83	95.67	
Baseline + GWKDC + TCPSAO (Proposed)	98.87	97.54	

The robustness of the proposed method was examined over six distinct imaging situations that replicate common issues encountered in agricultural settings. These encompassed Gaussian noise, low illumination, high contrast, image blurring, and previously unobserved test images from the same categories as the training pictures. Gaussian noise was introduced to replicate sensor-level distortion characterized by a normal distribution with a mean ( μ) of 0 and a standard deviation ( σ) of 0.01. Low light was emulated by reducing image brightness by 40%, while high contrast photos were generated by augmenting both brightness and contrast to replicate overexposed settings. To simulate motion or focus-based distortions, images were blurred using a Gaussian blur with a kernel size of k = 5. A distinct collection of unseen datasets, representing the same disease categories from the training dataset, was employed to assess the model’s generalization capability.

Figure 4 indicates that the model exhibited elevated values across all metrics, with negligible declines attributed to perturbations. Remarkably, even in challenging settings such as high contrast or input blurring, the model achieved F1-scores exceeding 90% and AUC values surpassing 94%. These performances confirm that the proposed methodology is both accurate under ideal circumstances and resilient under diverse and suboptimal input settings, hence verifying its suitability for real-world implementation.

Figure 4 Illustration the performance of the proposed model evaluated under six imaging conditions on both the Mendeley and Kaggle rice leaf disease datasets.

Comparison with state of the art methods

Table 12 gives a comparison of the various models on the amount of accuracy achieved on various datasets. Comparing the effectiveness of including custom CNN (Kazi & Palkar, 2024) and decision tree (Kulkarni & Shastri, 2024), it can be seen that both models yield an accuracy of 95.25% and 95.00%, respectively, on Kaggle. The model, by using K-means Segmentation with a custom CNN based on the classification model (Hasan et al., 2023) on Mendeley, achieves 92.70% accuracy, which is slightly lower than the other classification models. Most importantly, InceptionV3 was used for the classification of the Mendeley dataset, and similar accuracy was reported at 97.47% and 98.04% (Maulana et al., 2023; Firnando et al., 2024), proving the reliability of InceptionV3. An ensemble model comprising SqueezeNet was conducted using a combined dataset and achieved a fair accuracy of 93.30% (Kaur, Guleria & Trivedi, 2024). The proposed model is distinguishable for the highest accuracy of the forecast: 98.87% for Mendeley and 97.54% for Kaggle, which points out the high performance and universality of the estimates.

Table 12 Comparison of the proposed model with state-of-the-art methods.

Model	Year	Dataset	Accuracy (%)	
Custom CNN (Kazi & Palkar, 2024)	2024	Kaggle	95.25	
Decision tree (Kulkarni & Shastri, 2024)	2024	Kaggle	95.00	
K-mean segmentation and custom-CNN-based classification (Hasan et al., 2023)	2023	Mendeley	92.70	
InceptionV3 (Maulana et al., 2023)	2023	Mendeley	97.47	
InceptionV3 (Firnando et al., 2024)	2024	Mendeley	98.04	
Ensemble model with SqueezeNet (Kaur, Guleria & Trivedi, 2024)	2024	Combined (Mendeley + Kaggle)	93.30	
Proposed	–	Mendeley	98.87	
		Kaggle	97.54	

Conclusion

This study proposes a new way of categorizing rice leaf diseases, whereby the latest methods of preprocessing, segmentation, and optimization provide greater accuracy and efficiency. There were enhanced performances of 98.87% and 97.54% respectively, for the two datasets, and this shows how transfer learning can be applied to diagnose diseases in agriculture. These results clearly demonstrate that the proposed design is feasible for the segmentation and classification steps. In addition to its effective high classification performance, the proposed model is also engineered for deployment. With a model size of about 41 MB and an average inference time of 0.080 s per image, it is exceptionally suitable for real-time deployment on mobile or embedded devices. In the future, the endeavor will be broadened to diagnose diseases affecting numerous agricultural components, including branches, stems, and fruits. A mobile application will be developed and installed to segment and classify rice lesions, assisting farmers and professionals in identification. This program enables farmers to determine the health status of rice leaves, identifying whether they are healthy or afflicted by disease.

Supplemental Information

Supplemental Information 1 Comparison of Methods for Rice Leaf Disease Detection.

Supplemental Information 2 Sample images taken from Mendeley dataset.

Supplemental Information 3 Sample images taken from Kaggle dataset.

Supplemental Information 4 Architecture of EfficientNetB0.

Supplemental Information 5 Accuracy and loss graph for the proposed model on the Mendeley dataset.

Supplemental Information 6 Accuracy and loss graph for the proposed model on the Kaggle dataset.

Supplemental Information 7 Visual comparison of optimizers performance.

Additional Information and Declarations

Competing Interests

The authors declare that they have no competing interests.

Author Contributions

Samia Nawaz Yousafzai conceived and designed the experiments, performed the computation work, authored or reviewed drafts of the article, and approved the final draft.

Fahd N. Al-Wesabi conceived and designed the experiments, authored or reviewed drafts of the article, and approved the final draft.

Hadeel Alsolai performed the experiments, prepared figures and/or tables, and approved the final draft.

Shouki A. Ebad performed the experiments, analyzed the data, prepared figures and/or tables, and approved the final draft.

Inzamam Mashood Nasir performed the computation work, authored or reviewed drafts of the article, and approved the final draft.

Emad Fadhal analyzed the data, prepared figures and/or tables, and approved the final draft.

Adel Thaljaoui conceived and designed the experiments, prepared figures and/or tables, and approved the final draft.

Data Availability

The following information was supplied regarding data availability:

The Rice Leaf Disease Image Samples dataset are available at Mendeley: sethy, prabira Kumar (2020), “Rice Leaf Disease Image Samples”, Mendeley Data, V1, doi: 10.17632/fwcj7stb8r.1.

The Rice Leafs Disease Dataset is available at Kaggle: https://www.kaggle.com/datasets/dedeikhsandwisaputra/rice-leafs-disease-dataset.

The source code is available at Zenodo: Samia Nawaz. (2025). Samia-Nawaz/Segmentation-and-classification-of-rice-leaf-disease: v1.0.0 (v1.0.0). Zenodo. https://doi.org/10.5281/zenodo.15373402.

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
