# Peer review of "Advanced clustering and transfer learning based approach for rice leaf disease segmentation and classification"

_PeerJ Computer Science, doi:10.7717/peerj-cs.3018_

## Round 0.1 · original submission · Major Revisions

A reviewer suggests rejection, but the other reviewer considers your work slightly more positively. I suggest that you please revise the manuscript thoroughly, paying attention to both reviewers. English grammar and misspellings should be avoided.

**Language Note:** The Academic Editor has identified that the English language must be improved. PeerJ can provide language editing services - please contact us at [email protected] for pricing (be sure to provide your manuscript number and title). Alternatively, you should make your own arrangements to improve the language quality and provide details in your response letter. – PeerJ Staff

Reviewer 1 ·

Basic reporting

This study introduces a novel deep-learning framework that combines advanced image preprocessing, segmentation, and optimization techniques to accurately diagnose and classify rice leaf diseases. The framework utilizes a new clustering method called Gravity Weighted Kernelised Density Clustering (GWKDC) to segment diseased regions effectively, as well as a fine-tuned EfficientNetB0 model for deep feature extraction. To improve convergence and training efficiency, the study proposes an optimization algorithm—Tent Chaotic Particle Snow Ablation Optimizer (TCPSAO). The framework was evaluated on two public datasets (Mendeley and Kaggle), achieving classification accuracies of 98.87% and 97.54%, respectively. The proposed method outperformed other deep learning models and optimization techniques, proving its effectiveness and robustness in real-world agricultural applications.

Experimental design

The experimental design process is unclear because it is not explained why two separate data sets are used. In addition, the evaluation and organization of the experiment and the results do not compare the optimization of the method.

Validity of the findings

This finding is unclear when many segmentation and optimization methods have been compared on rice datasets. Still, it has not been thoroughly addressed in related studies or comparisons of methods.

Additional comments

However, there are several shortcomings in the current version that the authors should address:

1) The proposed approach combines optimization algorithms and segmentation methods. However, since both techniques are integrated in the experimental phase, it is unclear whether the performance improvements are due to the optimization algorithm or the segmentation process.

2) The comparison in Tables 2, 3, 4, 6, 7, and 8 is not entirely clear, as the experimental scenarios implemented by the authors are not well described, and the data are presented separately, making comparison difficult.

3) Although the model achieves high accuracy, important computational metrics such as training time, model size, and inference latency are not clearly reported.

4) The model is trained on two specific datasets. There is no evaluation of its robustness when applied to noisy images, low-light conditions, or previously unseen diseases.

5) The model is entirely trained and evaluated on pre-collected datasets (Kaggle and Mendeley). However, the authors have not explained why the two datasets are used separately or why they are not combined into a single dataset for training and testing.

6) In the Conclusion section, the authors mention plans to deploy the model on mobile devices in the future. However, they have not assessed the feasibility of such deployment based on the current study.

7) Both segmentation techniques and optimization algorithms have been widely studied and compared in previous research. Therefore, the authors should explore existing limitations in these areas and focus more clearly on addressing them in their study.

Reviewer 2 ·

Basic reporting

• This paper has an interesting contribution in rice leaf disease classification using clustering and transfer learning methods
• The introduction and literature review are quite clear in describing the background of the problem, the importance of the research, and the proposed method.
• The use of English in the paper is quite good academically

Experimental design

1. Add performance analysis of the segmentation method
2. Clarify the amount and distribution of training and test data
3. Provide a segmentation algorithm
4. Show visualization of the segmentation results to demonstrate the effectiveness of the clustering method

Validity of the findings

1. The results are impressive, especially the high AUC values, which indicate that the model is highly capable of distinguishing between diseased and healthy rice leaves.
2. It would be beneficial to include more detailed information on the model's performance under various conditions (e.g., varying image quality or environmental factors).

Additional comments

Correct any spelling errors, especially those related to equations and their explanations.

---

## Round 0.2 · accepted · Accept

Although one reviewer rejected the manuscript, it contains valuable information, and the authors have addressed all of the reviewers' comments to the best of my knowledge.